# Assessing local cultural awareness in university EFL learners: A Delphi and AHP-based index framework

**Lunan Fan**[1]*, **Guiping Cheng**[1], **Na Yan**[2]

1 Department of Theoretical Education, Guizhou Police College, Guiyang, Guizhou Province, China,
2 School of Foreign Languages, Guizhou Education University, Guiyang, Guizhou Province, China

* Fanlunan318@sina.com

## Abstract

To address the neglect of local cultural identity in university EFL learners' development, this study has developed and validated an evaluation framework for assessing their cultural awareness, with direct implications for curriculum reform. The framework operationalizes Kramsch's symbolic competence theory and Byram's ICC principles into measurable dimensions, achieved through two rounds of Delphi consultations with 15 EFL education experts (Kendall's W = 0.86, p < 0.01) and one round of Analytic Hierarchy Process (AHP) analysis (consistency ratios < 0.1). The finalized index system comprises three primary dimensions—Local Expression and Application (45.27%), Local Affective Attitudes (33.01%), and Local Cognition and Understanding (21.72%)—alongside 18 secondary indicators. Key innovations include balancing cultural pride and cross-cultural openness at identical weights (0.1181) and integration with CEFR/ACTFL via descriptor mapping (e.g., mapping local storytelling to CEFR C1 "pluricultural mediation"). This framework equips educators, curriculum developers, and policymakers with practical tools to design culturally grounded assessments and curricula, advancing a shift beyond Anglophone-centric models.

## Introduction

In today's globalized world, English proficiency functions as an essential skill for cross-border communication. Consequently, English as a Foreign Language (EFL) programs increasingly aim not only to cultivate linguistic competence but also to foster global engagement capabilities. However, learners frequently encounter "Cultural Aphasia" [1]—defined as the persistent difficulty in conveying native cultural nuances despite advanced foreign language proficiency. This linguistic-cultural disjuncture has catalyzed reforms in EFL curricula, where there is growing recognition of the importance of integrating localized content into language learning to make it more relevant and meaningful to learners' lived experiences [2,3].

**Data availability statement:** All data files are available from the Figshare database (link: https://figshare.com/s/d40943028c0d35d84f8f) with DOI: 10.6084/m9.figshare.29115743 under a CC BY 4.0 license.

**Funding:** This work was supported by the Guizhou Provincial Education Science Planning Project (Project No. 2023B035), entitled "Construction of National Consciousness in College English through Integration of Guizhou's Regional Cultural Resources" The funders had no role in study design, data collection and analysis, decision to publish, or preparation of the manuscript. Lunan Fan, the first author of this manuscript, is the recipient of the funding award listed above.

**Competing interests:** The authors have declared that no competing interests exist.

Although the advantages of integrating local elements into the EFL curriculum have gained more attention, it is still challenging to effectively assess how learners exercise their agency and demonstrate local identity through their plurilingual competence in English language education [4]. At present, most of the evaluation systems excessively focus on the assessment of language skills and cross-cultural communication ability. For example, frameworks such as the CEFR are effective in benchmarking grammatical accuracy. However, they inadequately capture learners' ability to articulate culturally – grounded narratives. This lack of assessment is particularly prominent in areas with significant cultural diversity, because the local cultural characteristics profoundly affect the actual application scenarios of language.

Given the importance of evaluating local cultural awareness in EFL learning, this study aims to develop a comprehensive evaluation index system. And the system will focus on three key dimensions: cognitive, affective, and behavioral. Such dimensions have been identified in previous research as critical components of language learning and intercultural competence [5,6]. In particular, the cognitive dimension will measure learners' understanding of local cultural, social, and historical contexts, and their ability to express this knowledge in English. The affective dimension will evaluate learners' attitudes toward expressing local content, including their emotional connection to their region and their motivation to share local stories. Finally, the behavioral dimension mainly focuses on how learners use English to express and spread local culture in practical communication.

The evaluation system constructed by this research will serve as a valuable resource for educators, enabling them to evaluate how well learners can interact with and represent their local culture when using English. The goal is to promote a comprehensive approach to language learning that integrates both global and local perspectives, fostering learners who can confidently and skillfully interact within diverse cultural contexts.

## Literature review

### Local cultural awareness in EFL learning

Local cultural awareness in EFL learning refers to the conscious integration of learners' local context into the language learning process. The idea of embedding local content in EFL curricula stems from the understanding that language learning should not be isolated from the socio-cultural background [7,8]. Grounded in sociocultural theory [9], local cultural consciousness emphasizes the importance of situational sensitivity. Educators need to be able to recognize and respond to the complex interactions within the sociocultural context in which learners are situated, such as language use, historical background, and community connections. Through utilizing cultural symbols and designing collaborative learning activities, educators can help students integrate theoretical knowledge with their local experiences, forming a meaningful knowledge network. This approach aims to foster the co-development of students' cognitive growth, cultural identity, and critical thinking skills. Local cultural awareness is conceptualized as a dynamic metacognitive skill [10] that enables learners

to "notice, analyze, and negotiate linguistic hybridity", such as code-switching patterns in multilingual communities. This construct extends beyond passive cultural literacy.

## Theoretical foundation: symbolic competence in local cultural expression

Kramsch's theory of symbolic competence [11] establishes the epistemological framework for local cultural awareness in EFL contexts. This theory redefines language learning as a dynamic process of meaning negotiation through historically embedded symbolic systems, contrasting sharply with transactional communication models [12].

Central to this perspective is the treatment of linguistic forms as semiotic resources that construct cultural identities rather than transmit information [12]. When EFL learners engage with English, they occupy a hybrid "third space" [13] where global and local narratives intersect, necessitating active navigation of one's identity through symbolic mediation. This process requires historicity—an awareness of how language preserves cultural memory and socio-political legacies that shape modern expression [11,13].

This framework directly explains the "cultural aphasia" phenomenon introduced earlier. Kramsch's lens interprets such communicative failures not as lexical deficits but as breakdowns in wielding English for cultural self-representation [11]. Consequently, assessing local cultural awareness must evaluate learners' ability to deploy L2 resources for authentic cultural identity expression—a dimension overlooked in standardized EFL assessments. Pedagogically, empirical evidence confirms that structured feedback effectively scaffolds this expressive ability [14]. This pedagogical intervention aligns with the naturalistic strategies of multilingual speakers, who incorporate their own linguistic resources to negotiate meaning and construct new norms [15], thereby providing a theoretical rationale for such classroom practices.

The theoretical foundation demands implementation through structured frameworks (Fig 1). Byram's Intercultural Communicative Competence (ICC) model [5] provides a compatible measurement architecture, categorizing competence into knowledge, attitudinal, and skill-based dimensions. While Kramsch's work establishes the theoretical necessity of L2-mediated identity construction, Byram's framework supplies the methodological tools for its evaluation. This integration offers dual benefits: Kramsch's focus on symbolic action deepens the conceptual basis of assessment, while Byram's dimensional taxonomy ensures operational feasibility. Theoretical alignment emerges through dimensional correlations: knowledge acquisition enables engagement with cultural historicity; attitudinal development reflects negotiated positioning in intercultural spaces; and skill demonstration facilitates performative praxis within third spaces.

## Cognitive and interdisciplinary demands in tertiary-level EFL education

English plays a vital role in cultural integration within foreign language education, making cultural inclusion in TEFL instruction critically important [16]. Contemporary university-level EFL teaching requires combining higher-order thinking (HOT) with cross-disciplinary cognitive skills. Advanced learners at this level must move beyond basic language mastery to demonstrate analytical abilities for processing complex, multi-subject content [17]. Integrating HOT into curricula pushes students past memorization, developing their critical analysis, problem-solving, and information synthesis across disciplines [3,10]. Learners tackle interdisciplinary challenges in English, demanding knowledge integration from multiple fields to form sophisticated responses [18]. This interplay of linguistic proficiency and cross-domain thinking is vividly illustrated by digital storytelling (DST) [19], which has shown cross-subject efficacy in EFL contexts: through weaving linguistic competencies, technical knowledge, and social identity expression into narrative creation, DST effectively enhances STEM literacy while embodying the HOT-driven, interdisciplinary learning paradigm.

Moreover, studies in Content and Language Integrated Learning (CLIL) confirm HOT and interdisciplinary skills' necessity for advanced EFL education. CLIL approaches emphasize the teaching of both content and language simultaneously. By encouraging learners to use English as a medium to acquire knowledge of subjects such as history, geography, and science, this approach strengthens both language proficiency and cognitive development [20,21]. Complex cognitive skills such as critical analysis, reflection, and synthesis are required, building competencies that correspond to modern educational goals.

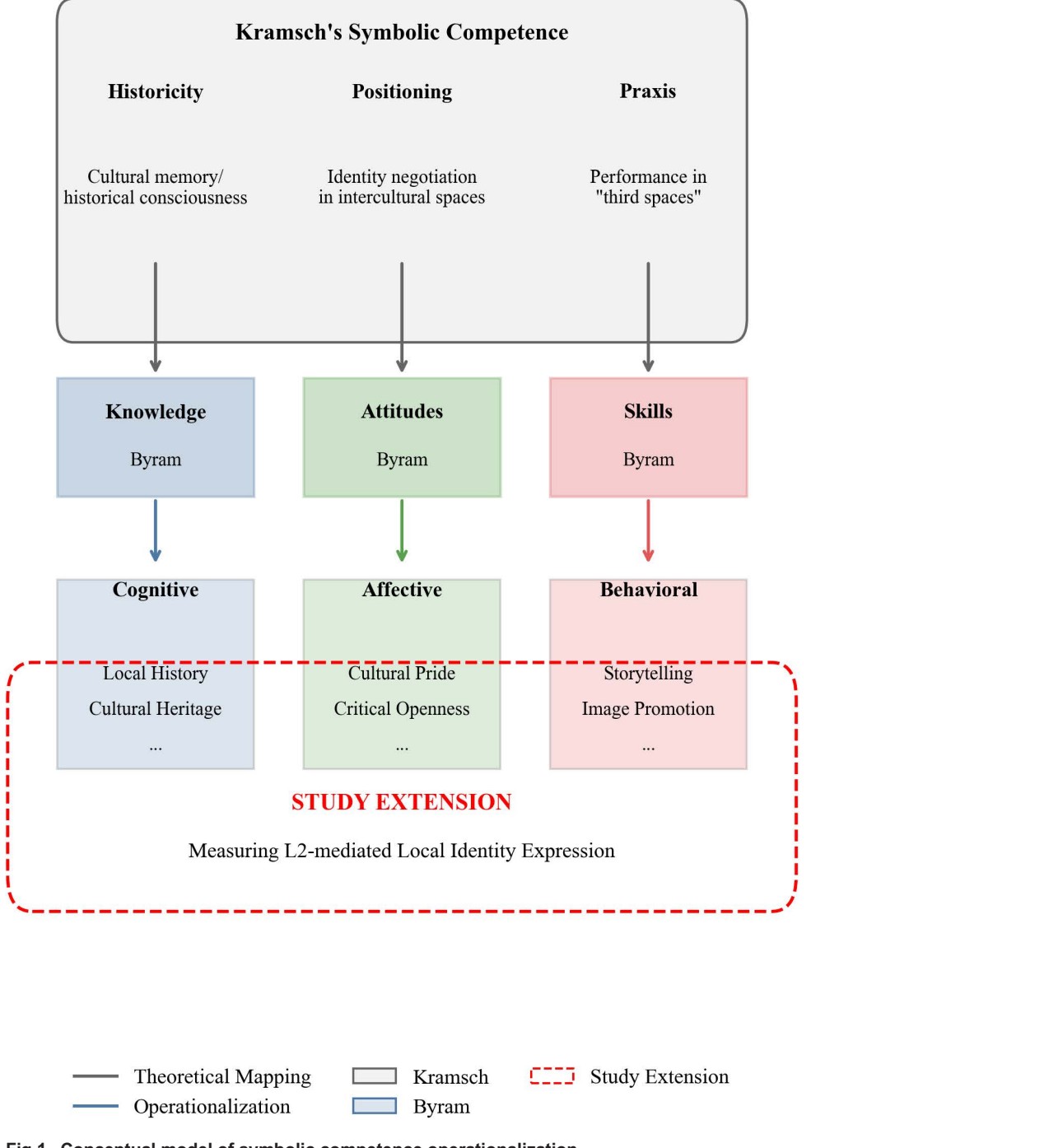

**Fig 1. Conceptual model of symbolic competence operationalization.**

## Benefits of EFL's attention to local cultural awareness

Despite the importance of nurturing local cultural awareness in language learning, EFL programs have traditionally prioritized the cultures of English-speaking countries—often overlooking the cultivation of learners' awareness of their own local contexts. This oversight can create a disconnect between learners and their cultural realities: for example, Indonesian

junior high textbooks, which emphasize Western cultural references while neglecting regional traditions [22], not only limit students' familiarity with their own customs but also weaken opportunities to develop local cultural awareness. Such imbalance, where foreign traditions overshadow local values in discussions, frames English as a tool for engaging with external cultures rather than a medium to reflect on and express one's own—potentially dampening motivation to connect language learning with personal cultural identity and hindering holistic acquisition [23].

Studies across diverse contexts (e.g., Thailand [24], Indonesia [25], China [26], Kosovo [27], and Argentina [3]) consistently highlight the benefits of EFL's focus on local cultural awareness, including enhanced student motivation, strengthened cultural pride, improved language skills, and deeper intercultural competence. For instance, when EFL instruction prioritizes local cultural awareness—such as through community-based content [24] or digital platforms that foster reflection on local contexts [23]—students show greater motivation [25,27], and local cultural narratives that cultivate this awareness have been shown to boost both cultural pride and language proficiency [3].

From a cognitive perspective, language and culture are fundamentally inseparable; language acquisition extends beyond mastering linguistic forms to encompass cultural meaning comprehension, thereby deepening cognitive processing [28]. This aligns with findings that the cognitive feedback provided by digital tools can significantly impact learners' cognitive development and application of cultural knowledge in language learning [29]. It is further demonstrated that task-induced involvement and time on task can significantly influence vocabulary acquisition in L2 learners, suggesting that culturally-relevant tasks enhance cognitive engagement and cultural knowledge integration [30]. Recent research reinforces this idea by highlighting how feedback mechanisms can enhance learners' cognitive engagement with cultural content, thereby enriching their understanding and application of cultural knowledge in language learning contexts [31]. Integrating local content also enhances intercultural competence and practical language skills [32]. Navigating local and global cultures via English strengthens communicative competence [15], particularly when learners engage with authentic cultural scenarios to develop practical communication abilities [33].

These findings confirm that effective EFL instruction requires balancing global communication needs with intentional attention to nurturing learners' local cultural awareness. By prioritizing this awareness, EFL not only bridges the gap between language learning and learners' cultural realities but also fosters more meaningful and holistic language development.

## Gaps in current assessment frameworks

Although integrating local content into EFL curricula shows clear benefits, measuring learners' cultural engagement through English remains challenging. Widely used evaluation frameworks, such as the Common European Framework of Reference for Languages (CEFR), focus primarily on assessing linguistic proficiency and intercultural competence but do not adequately address the assessment of local cultural awareness [34]. For example, the CEFR's "can do" descriptors primarily emphasize learners' ability to function in target-language contexts—for instance, "understanding main points of clear standard input on familiar matters (work, school, leisure)" (CEFR B1) [35]. While the framework includes mediation as a key communicative activity, defined as a socio-cultural practice that "promotes communication and collaboration" and "resolves sensitive issues" across languages and cultures [35], this concept is largely framed as a gap-bridging process. It positions learners as neutral facilitators or interpreters, yet it crucially lacks explicit descriptors to assess their ability to proactively express, explain, or defend the unique elements of their cultural identity in English. This omission therefore fails to capture the competence of cultural self-representation—a critical component of balanced intercultural communication [2].

This limitation extends to other widely adopted frameworks. The ACTFL Proficiency Guidelines [36], while emphasizing task-based performance in real-world contexts, primarily measures learners' ability to "describe local cultural practices" as a sub-skill of interpersonal communication, without establishing explicit criteria for evaluating the depth or authenticity of cultural identity expression [37,38]. Similarly, UNESCO's Global Citizenship Education Framework (2015), though

containing a socio-emotional domain that addresses "sense of belonging to local communities," lacks operationalized indicators for assessing how learners articulate their cultural roots through language [39].

Even frameworks explicitly incorporating cultural dimensions exhibit gaps. China's Standards of English Ability (CSE) mandate "introducing Chinese traditional culture in English" as a competence requirement, but its assessment rubrics remain anchored in linguistic accuracy rather than cultural authenticity [40]. The German National Educational Standards, while integrating intercultural competence as a curricular component, have not adequately addressed the assessment of local cultural awareness. Existing assessment frameworks predominantly focus on the comparison of cultural differences and the understanding of target cultures, neglecting the evaluation of learners' awareness and expression of their own local culture [41].

These gaps stem from two structural limitations in current frameworks. Firstly, most separate linguistic competence from cultural identity assessment. For instance, CEFR's pluricultural competence descriptors (e.g., "mediating cultural ambiguities") remain isolated from language production scales. Secondly, frameworks like CEFR measure adaptability to external cultures but fail to capture the dynamic process of reconstructing local identity through L2 [42].

Recent attempts to address these gaps through CEFR companion volumes show limited progress. While introducing "mediation scales" for cultural interpretation, they still lack descriptors for self-referential cultural expression. It has been critically noted [43] that the dominant frameworks reduce culture to a set of learnable behaviors rather than a lived identity that interacts with language acquisition.

To address this gap, this study aims to establish a comprehensive evaluation index system through a series of surveys using the Delphi method, allowing researchers to analyze the opinions of expert panels by collecting, combining and quantifying multiple opinions on complex issues. By creating a tailored assessment system, this study aims to contribute to the growing body of literature on English learning by providing educators with a tool to assess learners' ability to integrate into local culture through English. This framework will enable educators to assess not only language proficiency, but also the learners' ability to use English as a medium to express their local identity. Ultimately, the goal is to promote a more comprehensive approach to language learning that values both global and local perspectives and fosters learners who have the confidence and ability to navigate different cultural environments.

## Methods

### Participants

In accordance with the Delphi method guidelines [44] that suggest a sample size of 15–20 experts for consensus – driven studies, 15 EFL educators (Table 1) were invited to participate in the study to form the expert panel. The recruitment period for this study spanned from November 30, 2023, to March 1, 2024. These participants were recruited from a variety

Table 1. Demographic characteristics of expert participants (N = 15).

| Category | Subgroup | n | Category | Subgroup | n |
|---|---|---|---|---|---|
| Age | ≤35 years | 4 | Research experience[a] | Extensive | 9 |
| | 36-45 years | 9 | | Moderate | 5 |
| | 46-55 years | 2 | | Limited | 1 |
| Years of EFL teaching experience | 10-19 years | 13 | Teaching experience[b] | Extensive | 11 |
| | 20-30 years | 2 | | Moderate | 4 |
| Highest degree | Ph.D. | 13 | Current teaching position | Full Professor | 10 |
| | Master's | 2 | | Associate Professor | 5 |

a Research experience criteria: Extensive (≥10 publications), Moderate (5–9), Limited (5).

b Teaching experience criteria: Extensive (≥10 years), Moderate (5–9), Limited (5).

of academic institutions across China, ensuring geographical and institutional diversity. All participants provided written informed consent. All participants provided written informed consent, and the study was approved by the Institutional Review Board of GPC College (GZPC20231127). To ensure the maintenance of expert anonymity during the Delphi rounds, the survey materials explicitly guaranteed strict confidentiality. Experts were not made aware of each other's identities, and all responses were collected and handled to ensure full anonymization throughout the process.

The expert panel was composed of EFL specialists who were required to meet the following three selection criteria: (1) a minimum of 10 years' experience in EFL curriculum design; (2) having conducted research in the field of EFL; and (3) holding senior professional titles such as Professor or Associate Professor. This rigorous selection process ensured that the panel's input was both authoritative and relevant to the study's objectives.

## Preliminary establishment of indicators

To comprehensively and systematically assess the awareness of native culture among advanced foreign language learners, this study adopted a multi-method approach to construct an initial set of indicators. First, researchers searched the Web of Science database using keywords such as "foreign language native awareness," "cultural competence assessment," and "place-based pedagogy." This search aimed to identify existing literature relevant to the topic, providing a solid theoretical foundation for the establishment of indicators. Concurrently, the research team (all front-line university EFL teachers) held two rounds of internal discussions to refine the initial indicator set by clarifying concepts, checking for redundancy, ensuring domain coverage, and verifying operational feasibility. Following this, a separate panel of 15 external EFL education experts was invited to participate in the formal Delphi consultation process. These steps were crucial for refining the initial set of indicators, ensuring that the selected indicators were not only conceptually sound but also practically feasible and applicable in the context of assessing the awareness of native culture among advanced foreign language learners.

The assessment framework was grounded in a Kramsch-informed adaptation of Byram's tripartite ICC model. The Kramsch-Byram integration achieves theoretical synergy: Byram's measurable dimensions are conceptually enriched through Kramsch's historicity-positioning-praxis triad [11,38]. Pedagogical utility further justifies this synthesis, as the combined model captures internal development (manifested through cultural consciousness informed by historicity) and external manifestations (realized via behavioral praxis in third spaces) [45]. Specifically, the cognitive dimension (codifying historicity) measures knowledge articulation of sociohistorical contexts through cultural schema activation [2,46]. The affective dimension (reflective positioning) evaluates attitudes toward cultural hybridity, including motivational investment in identity expression [47]. The behavioral dimension (applied praxis) assesses contextually adaptive communication that conforms to local linguistic norms [48].

This framework generated 3 primary indicators with 24 secondary sub-indicators for initial Delphi validation (S1 Table).

## Procedures

Under the theoretical guidance of the Kramsch-Byram integrated framework (Fig 2), this study conducted three rounds of expert consultations using paper-based inquiry materials to develop and validate cultural awareness indicators. The first two rounds conformed to the Delphi procedure, while the final round adopted the Analytic Hierarchy Process.
**Delphi procedure.** The Delphi method was used to gather expert opinions on the key indicators for assessing local cultural awareness in EFL learners. The method involved organizing several rounds of questionnaires to solicit feedback from experts in the field of language education. Through iterative surveys and feedback processes, consensus on specific evaluation indicators was sought. Anonymity in the process allowed for more candid expert feedback, while the statistical analysis provided insight into the central tendencies and variances in expert opinions. These characteristics make the Delphi method particularly effective for educational research where diverse expert perspectives are required [44].

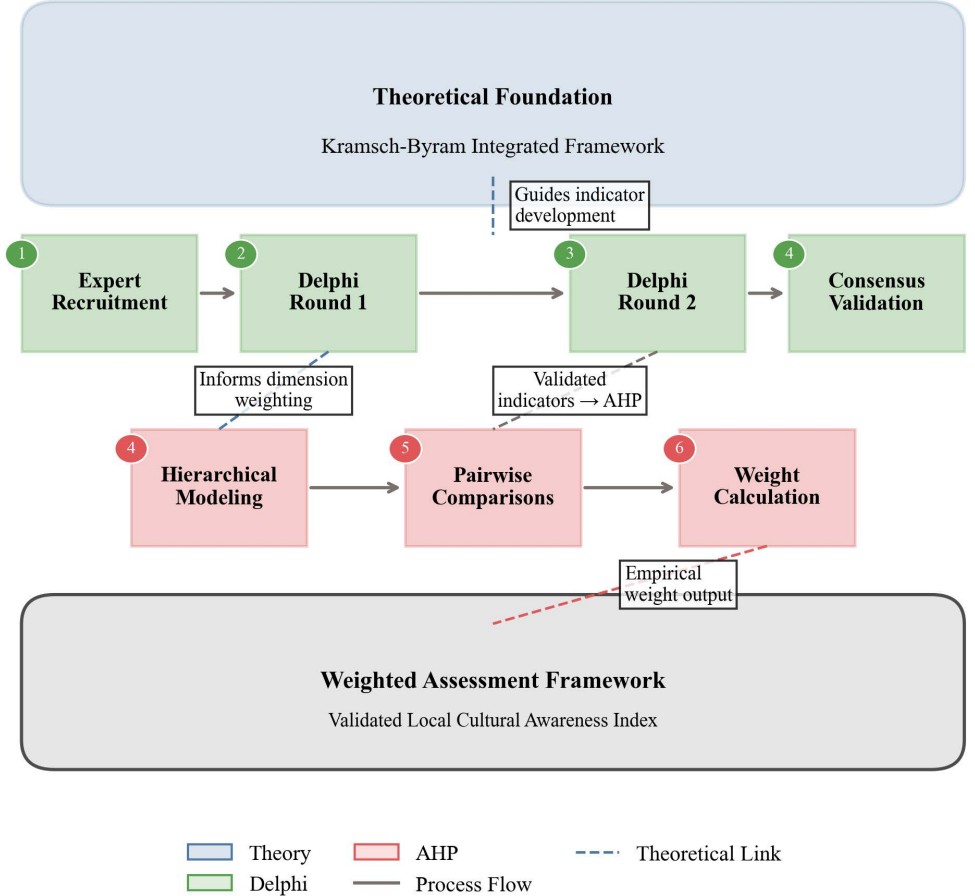

**Fig 2. Methodological workflow integrating theoretical frameworks with Delphi-AHP procedures.**

In the first round, opinions were collected from experts on the necessity of the three primary indicators and 24 secondary indicators. The analysis involved descriptive statistics, including the calculation of the mean (M) and standard deviation (SD) to validate initial indicators. This survey lasted for about 4 weeks, during which experts were asked to use a 5-point scale to evaluate each indicator, ranging from "not important" to "very important". In this round of evaluation, participants also rated their familiarity with evaluation criteria (Cs) and described their judgment rationale (Ca). The basis for judgment (Ca) includes practical experience, logical reasoning, and an understanding of domestic and international language education trends. These self-assessments help ensure that the feedback provided is based on a well-informed perspective. The formula for calculating the expert authority coefficient Cr is:

$$Cr = \frac{Ca + Cs}{2}$$

It is generally believed that when the expert authority coefficient Cr is greater than or equal to 0.7 [49], the research results are considered reliable. The assignments for Ca and Cs are shown in Table 2 and Table 3 respectively.

The second round focused on revising and refining the indicators based on the feedback from the first round. A revised version of the assessment system was developed for this round. After collecting and analyzing the feedback from the first round, indicators considered irrelevant or meaningless were removed. Based on the degree of consensus among experts

**Table 2. Assignment for expert judgment basis (Ca).**

| Judgment Basis | Impact Level | | |
|---|---|---|---|
| | High Impact | Moderate Impact | Low Impact |
| Deductive Reasoning | 0.3 | 0.2 | 0.1 |
| Practical Experience | 0.5 | 0.4 | 0.3 |
| Understanding of Domestic and International Language Education Trends | 0.1 | 0.1 | 0.1 |
| Intuition | 0.1 | 0.1 | 0.1 |
| Total | 1.0 | 0.7 | 0.6 |

**Table 3. Familiarity coefficient assignment (Cs).**

| Familiarity Level | Coefficient (Cs) |
|---|---|
| Extremely Familiar | 1.0 |
| Very Familiar | 0.75 |
| Moderately Familiar | 0.50 |
| Slightly Familiar | 0.25 |
| Not Familiar | 0.00 |

of expert opinions, researchers re-sorted the remaining indicators and revised the content of the consultation questionnaire to ensure its accuracy and applicability. Indicators with low consensus (M < 3.5, SD > 1.0, CV > 25%) were eliminated to ensure a higher level of agreement among experts.

**Analytic Hierarchy Process (AHP).** The final round was dedicated to confirming consensus among experts and constructing the Analytic Hierarchy Process (AHP) hierarchy. The AHP was employed to assign weights to the various indicators of local cultural awareness based on expert judgment. By breaking down the complex task of evaluating learners' local cultural awareness into multiple hierarchical levels, AHP facilitated precise pairwise comparisons which helped determine the relative importance of each indicator within the overall evaluation framework. The use of the Saaty 1~9 scale [50] in AHP provided a structured method for experts to rate the importance of each indicator, ensuring that the weight distribution reflected the collective expert judgment.

To ensure the reliability of the pairwise comparisons, the consistency ratio (CR) was calculated using the formula CR = CI/RI, where CI is the consistency index and RI is the random consistency index. According to established criteria, a CR value less than 0.1 indicates acceptable consistency [50]. This verification step ensured that the judgments made by experts are logically consistent and free from significant biases or errors.

## Results

### Expert authority analysis

As shown in Table 1, the participants all had a high educational background, 87% held doctoral degrees, and they had a mean of 15.9 years of experience (SD = 4.01). They also had rich experience in university foreign language teaching and good professional representation, which meant they could provide valuable information on the structural rationality of the index system.

In the three rounds of discussion and consultation, all 15 experts participated in the entire process, resulting in a 100% questionnaire recovery rate. This demonstrated the high participation and engagement of the experts and provided strong support for the progress of the research. The experts also conducted a self-assessment on their basis for judgment and their familiarity with the topics. After calculation, the expert authority coefficient (Cr) was 0.867, indicating that the experts

had a high degree of familiarity with the indicators as well as high authority in their judgments. Thus, the consultation results have practical reference significance [47]. The authority coefficient of the experts is shown in Table 4.

## Consensus development analysis

To assess the degree of agreement among the experts, the Kendall coefficient of concordance (W) was calculated. This non-parametric statistic measures the consistency of a group of judges or raters on the items they ranked [51]. The Kendall's W coefficient and chi-square tests revealed significant consensus improvement across Delphi rounds (Table 5).

The initial round yielded moderate consensus (W = 0.634), which is typical during the early exploratory stage of indicator generation. Moreover, the significant $\chi^2$ value (p < 0.001) suggested non-random agreement, justifying further refinement of the indicators [52].

The second round of consultation resulted in a substantially higher Kendall's W value of 0.855 (p < 0.001). Although this value slightly fell outside the conventional range of 0.6–0.8 typically indicative of strong consensus [53], it reflects an

**Table 4. Authority coefficients of 15 experts.**

| Expert Number | Familiarity Coefficient (Cs) | Basis of Judgment (Ca) | Authority Coefficient (Cr) |
|---|---|---|---|
| 1 | 1.00 | 1.00 | 1.000 |
| 2 | 0.75 | 0.90 | 0.825 |
| 3 | 0.75 | 1.00 | 0.875 |
| 4 | 1.00 | 0.90 | 0.950 |
| 5 | 1.00 | 0.70 | 0.850 |
| 6 | 0.50 | 0.90 | 0.700 |
| 7 | 1.00 | 0.70 | 0.850 |
| 8 | 0.75 | 1.00 | 0.875 |
| 9 | 0.75 | 0.90 | 0.825 |
| 10 | 1.00 | 1.00 | 1.000 |
| 11 | 0.75 | 0.70 | 0.725 |
| 12 | 0.75 | 1.00 | 0.875 |
| 13 | 0.75 | 1.00 | 0.875 |
| 14 | 1.00 | 0.90 | 0.950 |
| 15 | 0.75 | 0.90 | 0.825 |
| Mean | 0.83 | 0.90 | 0.867 |

Notes:

1. Cr is calculated as Cr=(Cs+Ca)/2.

2. The mean Cr value is derived from precise calculations (retaining four decimal places) before rounding.

3. A Cr value ≥ 0.7 indicates high reliability of expert opinions.

**Table 5. Expert consensus analysis across Delphi rounds.**

| Round | Experts (n) | Indicators (k) | Kendall's W | $\chi^2$(df) | p-value | Interpretation |
|---|---|---|---|---|---|---|
| 1 | 15 | 24 | 0.634 | 267.63 (23) | <0.001 | Moderate consensus |
| 2 | 15 | 19* | 0.855 | 217.9 (18) | <0.001 | Strong consensus |

Notes:

1. *5 indicators eliminated after Round 1 screening.

2. Degrees of freedom (df) calculated as df = k − 1.

exceptional level of agreement among experts. The 34.8% increase from Round 1 to Round 2 demonstrates the effectiveness of the iterative Delphi process in harmonizing diverse expert perspectives and achieving robust consensus.

### First Delphi survey results

The initial Delphi round evaluated 24 secondary indicators across three primary dimensions. By applying consensus thresholds (M ≥ 3.5, SD < 1, CV < 0.25), 17 indicators were retained (Table 6), while 7 were eliminated or revised due to low relevance or high divergence (Table 7).

Specifically, *Local Political System and Governance* and *Participation in Culturally Relevant Discussions* were excluded because of their measurement challenges. Experts recommended consolidating the indicators addressing economic, social, and global issues into a unified construct—*Local and Global Issues in Everyday Contexts*—to prioritize practical applications, while environmental topics were integrated into Local Development Achievements. Furthermore, *Cognition of National and Regional Geography* was renamed *Local Scenic Beauty* to strengthen its cultural linkages. The descriptors for historical and ethical indicators were refined to enhance their operational alignment with EFL objectives. The revised framework, which comprised 18 indicators, proceeded to the second Delphi round.

### Second Delphi survey results

The second Delphi survey (S2 Table) assessed expert consensus on the appropriateness of 18 indicators. All indicators met the retention criteria (M ≥ 3.5, SD < 1, CV < 0.25), with mean values ranging from 3.93 to 5.00, standard deviations ranging from 0.00 to 0.45, and coefficients of variation ranging from 0.08 to 0.09, which confirmed high agreement among experts. The results are presented below, organized by dimension.

**Local cognition and understanding.** Experts exhibited strong consensus on indicators within this dimension (Table 8). *Local Cultural Practices*, *Art, and Literature,* and *National Virtues and Qualities* received unanimous endorsement (M = 5.00, SD = 0.00); *Daily Life Experiences* and *Local Development Achievements* demonstrated near-perfect scores (M ≥ 4.93, CV < 0.05), underscoring their efficacy in bridging cultural abstractions with tangible contexts relevant to EFL

**Table 6. Retained indicators after first Delphi round (n = 17).**

| Indicator Name | Mean (M) | SD | CV |
|---|---|---|---|
| Local History | 4.47 | 0.74 | 0.17 |
| Local Ethical and Legal Systems | 4.07 | 0.59 | 0.15 |
| Regional Geography | 3.87 | 0.83 | 0.22 |
| Local Cultural Practices, Art, and Literature | 4.80 | 0.56 | 0.12 |
| Local Development Achievements | 4.67 | 0.62 | 0.13 |
| National Virtues and Qualities | 4.80 | 0.41 | 0.09 |
| Daily Life Experiences | 4.67 | 0.49 | 0.10 |
| Local Language and Dialect Varieties | 3.87 | 0.52 | 0.13 |
| Motivation to Express and Communicate Local Identity | 4.80 | 0.41 | 0.09 |
| Emotional Attachment to Local Culture | 4.13 | 0.64 | 0.15 |
| Cultural Pride | 5.00 | 0.00 | 0.00 |
| Openness in Cross-Cultural Engagement | 5.00 | 0.00 | 0.00 |
| Cultural Comparison | 4.00 | 0.65 | 0.16 |
| Interdisciplinary Knowledge Application | 4.67 | 0.49 | 0.10 |
| Adaptation of Language to Reflect Local Norms | 4.07 | 0.88 | 0.22 |
| Use of English for Local Storytelling | 5.00 | 0.00 | 0.00 |
| Promotion and Preservation of Local Image | 4.60 | 0.63 | 0.14 |

**Table 7. Unretained/ modified indicators after first Delphi round (n = 7).**

| Indicator Name | Mean (M) | SD | CV | Action Taken |
|---|---|---|---|---|
| Local Political System and Governance | 1.40 | 0.63 | 0.45 | Removed |
| Local Economic System | 3.07 | 0.70 | 0.23 | Removed |
| Local Social Issues | 2.88 | 1.06 | 0.37 | Integrated into a new indicator |
| International Relations and Global Issues | 2.93 | 0.80 | 0.27 | Removed |
| Local Educational System | 3.13 | 1.25 | 0.40 | Removed |
| Local Environment and Sustainable Development | 3.00 | 0.93 | 0.31 | Merged into A4 Local Development Achievements |
| Participation in Culturally Relevant Discussions | 2.27 | 0.70 | 0.31 | Removed |

**Table 8. Two-round Delphi survey results for secondary indicators under the primary indicator *Local Cognition and Understanding*.**

| Secondary Indicator | Mean (out of 5) | | SD | | CV | | Action Taken |
|---|---|---|---|---|---|---|---|
| | 1st | 2nd | 1st | 2nd | 1st | 2nd | |
| Local History | 4.47 | 4.93 | 0.74 | 0.26 | 0.17 | 0.05 | Retained |
| Local Scenic Beauty | 3.87 | 3.93 | 0.83 | 0.26 | 0.22 | 0.07 | Retained |
| Local Cultural Practices, Art, and Literature | 4.80 | 5.00 | 0.56 | 0.00 | 0.12 | 0.00 | Retained |
| Local Development Achievements | 4.67 | 4.93 | 0.62 | 0.26 | 0.13 | 0.05 | Retained |
| National Virtues and Qualities | 4.80 | 5.00 | 0.41 | 0.00 | 0.09 | 0.00 | Retained |
| Daily Life Experiences | 4.67 | 5.00 | 0.49 | 0.00 | 0.10 | 0.00 | Retained |
| Local Language and Dialect Varieties | 3.87 | 3.93 | 0.52 | 0.26 | 0.13 | 0.07 | Retained |
| Local Ethical and Legal Systems | 4.07 | 4.73 | 0.59 | 0.46 | 0.15 | 0.10 | Retained |
| Local and Global Issues in Everyday Contexts | – | 3.93 | – | 0.26 | – | 0.07 | Retained |

pedagogy. *Local History* achieved substantially stronger consensus in Round 2 (M = 4.93 vs. 4.47 in Round 1), reflecting greater conceptual clarity after the refinement of its descriptors. Similarly, *Local Ethical and Legal Systems* showed notable agreement improvement (M = 4.73 vs. 4.07), confirming enhanced operational viability. The newly introduced *Local and Global Issues in Everyday Contexts* (M = 3.93) met all retention thresholds despite its comparatively lower mean score. It was retained in the framework, validating its role in fostering critically engaged cultural awareness that is pertinent to university-level EFL curricula.

**Local affective attitudes.** All affective indicators received strong endorsement (Table 9). Notably, *Cultural Pride* and *Critical Openness in Cross-Cultural Engagement* achieved perfect consensus in both rounds, reflecting unanimous agreement on their critical role in fostering cultural confidence and balanced intercultural perspectives. Emotional Attachment to Local Culture retained high reliability (M = 4.00, CV = 0.09) despite a marginal decrease from Round 1 (M = 4.13), underscoring its stability in measuring learners' cultural belonging.

**Table 9. Two-round Delphi survey results for secondary indicators under the primary indicator *Local Affective Attitudes*.**

| Secondary Indicator | Mean (out of 5) | | SD | | CV | | Action Taken |
|---|---|---|---|---|---|---|---|
| | 1st | 2nd | 1st | 2nd | 1st | 2nd | |
| Motivation to Express and Communicate Local Identity | 4.80 | 4.93 | 0.41 | 0.26 | 0.09 | 0.05 | Retained |
| Emotional Attachment to Local Culture | 4.13 | 4.00 | 0.64 | 0.38 | 0.15 | 0.09 | Retained |
| Cultural Pride | 5.00 | 5.00 | 0.00 | 0.00 | 0.00 | 0.00 | Retained |
| Critical Openness in Cross-Cultural Engagement | 5.00 | 5.00 | 0.00 | 0.00 | 0.00 | 0.00 | Retained |

**Local expression and application.** Table 10 summarizes results of this dimension, demonstrating strong expert consensus on their relevance to assessing learners' ability to express and apply local culture through English. Two indicators—*Use of English for Local Storytelling* and *Promotion and Preservation of Local Image*—received unanimous support (M = 5.00, SD = 0.00), emphasizing storytelling and advocacy as critical skills for cultural dissemination. Minor variations among secondary indicators (e.g., Cultural Comparison M = 4.07) remained within consensus thresholds, collectively validating their role in transforming cultural knowledge into performative language practices essential for advanced EFL learners.

The unanimous validation of all 18 indicators confirms the theoretical coherence and pedagogical utility of the proposed framework. Its clear cognitive-affective-behavioral structure effectively operationalizes the construct of symbolic competence for EFL assessment. This strong consensus provides a robust foundation for the subsequent Analytic Hierarchy Process (AHP) to determine the relative weights of these indicators.

### Analytic hierarchy process (AHP) weighting outcomes

The third consultation phase employed the Analytic Hierarchy Process (AHP) to determine the relative weights of indicators within the evaluation system. Following AHP protocol, a hierarchical model (S1 Fig) was constructed based on the finalized primary and secondary indicators of EFL Learners' Local Cultural Awareness. Fifteen experts from the Delphi panel were invited to complete pairwise comparison matrices for the hierarchical structure (S3 Table, S4 Table).

The expert evaluations were analyzed using AHP software to calculate eigenvectors, perform hierarchical single ranking, and conduct consistency checks. All four judgment matrices exhibited satisfactory consistency, with consistency ratios (CR) below the threshold of 0.1 and consistency indices (CI) approaching zero (Table 11). These results confirm the logical coherence of expert judgments and validate the weight assignments.

Final weights for primary and secondary indicators are presented in Table 12 and visualized in Fig 3-Fig 4. Complete specifications of all indicators, including their operational definitions, are detailed in S5 Table. Among the primary dimensions, *Expression and Application* emerged as the dominant one with a weight of 0.4527, followed by *Local Affective Attitudes* (0.3301) and *Local Cognition and Understanding* (0.2172).

**Table 10. Two-round Delphi survey results for secondary indicators under the primary indicator *Local Expression and Application*.**

| Secondary Indicator | Mean (out of 5) | | SD | | CV | | Action Taken |
|---|---|---|---|---|---|---|---|
| | 1st | 2nd | 1st | 2nd | 1st | 2nd | |
| Cultural Comparison | 4.00 | 4.07 | 0.59 | 0.26 | 0.16 | 0.06 | Retained |
| Interdisciplinary Knowledge Application | 4.67 | 4.93 | 0.49 | 0.26 | 0.10 | 0.05 | Retained |
| Adaptation of Language to Reflect Local Norms | 4.07 | 4.07 | 0.88 | 0.26 | 0.22 | 0.06 | Retained |
| Use of English for Local Storytelling | 5.00 | 5.00 | 0.00 | 0.00 | 0.00 | 0.00 | Retained |
| Promotion and Preservation of Local Image | 4.60 | 5.00 | 0.63 | 0.00 | 0.14 | 0.00 | Retained |

**Table 11. AHP consistency metrics.**

| Hierarchy Level | *λmax* | CI | CR |
|---|---|---|---|
| Primary Dimensions | 3.1008 | 0.0504 | 0.0970 |
| Expression Sub-criteria | 5.3500 | 0.0875 | 0.0781 |
| Affective Sub-criteria | 4.2364 | 0.0788 | 0.0886 |
| Cognitive Sub-criteria | 9.3900 | 0.0488 | 0.0334 |

**Table 12. Hierarchical evaluation index system with AHP weight allocation for EFL Learners' Local Cultural Awareness.**

| Primary indicators | Secondary indicators | Weight | Combined weight |
|---|---|---|---|
| I-1Local Cognition and Understanding (0.2172) | II-1Local History | 0.1544 | 0.0335 |
| | II -2 Local Cultural Practices, Art, and Literature | 0.1544 | 0.0335 |
| | II -3 Local Development Achievements | 0.1544 | 0.0335 |
| | II-4 Daily Life Experiences | 0.1544 | 0.0335 |
| | II -5 National Virtues and Qualities | 0.1524 | 0.0331 |
| | II -6 Local Ethical and Legal Systems | 0.0921 | 0.0200 |
| | II -7 Local Scenic Beauty | 0.0510 | 0.0111 |
| | II -8 Local and Global Issues in Everyday Contexts | 0.0438 | 0.0095 |
| | II -9 Local Language and Dialect Varieties | 0.0430 | 0.0093 |
| I-2Local Affective Attitudes (0.3301) | II -10 Cultural Pride | 0.3578 | 0.1181 |
| | II -11 Openness in Cross-Cultural Understanding | 0.3578 | 0.1181 |
| | II -12 Motivation to Express and Communicate Local Identity | 0.1729 | 0.0571 |
| | II -13 Emotional Attachment to Local Culture | 0.1115 | 0.0368 |
| I-3 Local Expression and Application (0.4527) | II -14 Use of English for Local Storytelling | 0.3019 | 0.1367 |
| | II -15 Promotion and Preservation of Local Image | 0.3019 | 0.1367 |
| | II -16 Interdisciplinary Knowledge Application | 0.1843 | 0.0834 |
| | II -17 Adaptation of Language to Reflect Local Norms | 0.1178 | 0.0533 |
| | II -18 Cultural Comparison | 0.0942 | 0.0426 |

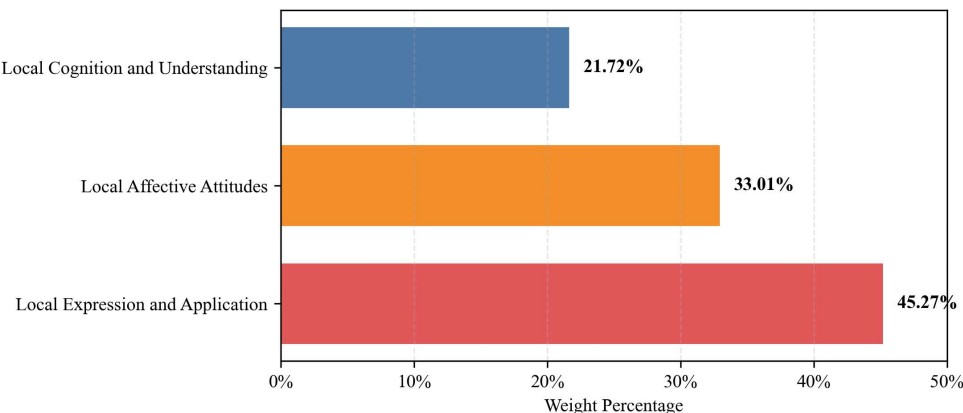

**Fig 3. Weights of primary indicators for local cultural awareness assessment.**

At the secondary level (Fig 4), *Use of English for Local Storytelling* and *Promotion and Preservation of Local Image* received the highest global weights, emphasizing narrative-driven cultural advocacy as a core skill. *Cultural Pride* and *Critical Openness in Cross-Cultural Engagement* dominated the dimension of *Local Affective Attitudes,* reflecting experts' emphasis on balancing cultural confidence. *Local History*, *Local Cultural Practices, Art, and Literature,* and *Daily Life Experiences* held equal weights, indicating their foundational equivalence in cultural knowledge acquisition.

## Discussion

This study constructed an evaluation index system for local cultural awareness in EFL learning through the Delphi method and Analytic Hierarchy Process (AHP), to address the underrepresentation of learners' native cultural identity. The

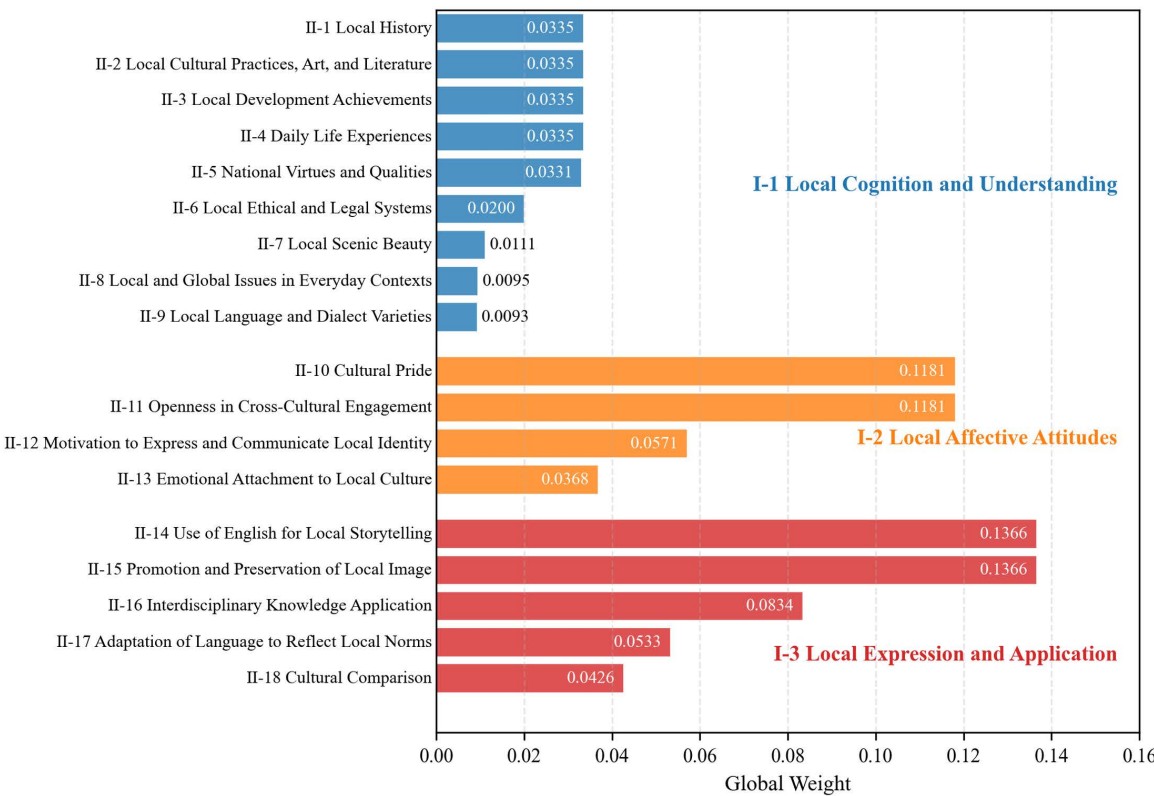

**Fig 4. Secondary indicator weights in cultural awareness assessment.**

discussion interprets the findings from educational and theoretical perspectives, linking the revision or retention of indicators to their practical significance in university EFL teaching.

The AHP-derived weights revealed that *Local Expression and Application* holds the highest priority, followed by *Local Affective Attitudes* and *Local Cognition and Understanding*. Among secondary indicators *Use of English for Local Storytelling, Promotion and Preservation of Local Image* emerged as the most critical skills, while *Cultural Pride* and *Critical Openness in Cross-Cultural Engagement* dominated the affective dimension. Meanwhile, *Local History*, *Local Cultural Practices, Art, and Literature*, *Local Development Achievements*, and *Daily Life Experiences* had relatively high weights under the dimension of Cognition, revealing the multifaceted emphasis on local cognition and understanding in English learning.

## Operationalizing and extending the Kramsch-Byram Synthesis

The empirical findings and indicator weights operationalize and extend the integrated Kramsch-Byram framework (Fig 2-Fig 3). Kramsch's symbolic competence (see Literature Review) positions language as a semiotic resource for meaning negotiation, identity construction, and navigating the "third space" within cultural historicity, while Byram's ICC model supplies the structural assessment dimensions.

Crucially, the index system concretizes Kramsch's symbolic action tenets—particularly active negotiation and inventive meaning construction within the third space—into specific dimensions, indicators, and assessment tasks. The framework further advances Kramsch's conception by integrating affective drivers, emphasizing native culture application, linking local symbols to global discourse, and providing pedagogical specificity for implementation.

## Local expression and application dimension

Local Expression and Application (0.453) was accorded the greatest relative importance, most directly realizing Kramsch's vision of learners actively creating meaning and reframing discourses through language. It moves beyond passive understanding to the active, creative deployment of English as a symbolic tool for expressing and negotiating local cultural identity. This finding is empirically supported by studies showing that action-oriented tasks enhance cultural retention [14].

By establishing a mechanism for applying ICC principles in context-specific language use, the index system bridges a critical gap in the existing literature by effectively translating theoretical frameworks into actionable pedagogy. This addresses a critical yet underexplored dimension in existing literature. For instance, students could document local heritage sites (e.g., ancient bridges with donation inscriptions) in English and compose analytical narratives to systematically decode the cultural meanings embedded in these spaces (see S5 Table). Through their writing, learners should illuminate how these inscriptions exemplify traditional values of fraternal cooperation and contractual ethics, bridging historical artifacts with contemporary understanding. Through the knowledge-action transformation mechanism, this design directly responds to the highest AHP-assigned weight of this dimension, translating the ICC model from a theoretical framework into teaching practice.

The secondary indicator of Use of English for Local Storytelling (II-14) holds one of the highest global weights. Learners are not merely translating stories; they are selecting culturally significant symbols (events, figures, legends), constructing a narrative framework in English, and infusing the narrative with cultural meaning. The higher weighting of Use of English for Local Storytelling (13.67%) compared to Local History (3.35%) is consistent with Kramsch's symbolic competence framework, which prioritizes active meaning negotiation over passive knowledge acquisition. Empirical pilot studies [3,18,19,23] have shown that narrative-based tasks significantly enhance learners' ability to mediate cultural identity in English, as they require selecting, interpreting, and reconstructing cultural symbols—skills critical for intercultural communication. In contrast, Local History (a component of cognitive understanding) was assigned lower weight because, while experts (via Delphi rounds) acknowledged its fundamental importance to EFL learning (e.g., providing cultural context for language use), the AHP process prioritized behavioral dimensions due to their more direct and frequent activation in real-world intercultural interactions—where narrative practices often serve as the primary vehicle for mediating cultural identity. This also resonates with Byram's ICC model, which emphasizes behavioral application over rote factual recall.

In university EFL courses, this indicator can be applied through collaborative projects: advanced learners may document entrepreneurial histories of local time-honored brands, producing podcasts that weave entrepreneurs' interviews with cultural analyses (II-14 in S5 Table). The narrative-driven approach diverges from earlier target-culture immersion studies [5,11], and recent work framing narratives as grammatical tools [54], addressing critiques of task-based methods that neglect cultural agency [15]. In contrast to the focus on refining cultural expression via corrective feedback [14], the index system foregrounds proactive cultural production. It embodies Kramsch's position that narrative empowers learners to explore and demonstrate symbolic power in a foreign language.

Similarly, the highly endorsed indicator of *Promotion and Preservation of Local Image* (II-15) echoes and strengthens scholarly calls for EFL pedagogy to enhance the application of cultural knowledge [3,14]. This sub-indicator embodies the critical and proactive dimensions of symbolic competence. Learners must first understand potential misinterpretations or negative symbolic representations, then strategically reframe the discourse using English symbols to clarify, defend, and reconstruct a more authentic or positive image. The example assessment criteria assess: (1) Expression Logic (rigorous argumentation), and (2) Cross-Cultural Communication Effect (audience adaptation). This directly assesses learners' capacity for meaning negotiation and cultural symbolic intervention, reflecting Kramsch's core principles. By designing tasks where students actively counter misrepresentations (II-15) Kramsch's theory is extended into the practical realm of cultural advocacy and discursive power-building within EFL.

## Emotional dimension

In the emotional dimension, the study shows balanced emphasis on *Cultural Pride* (II-10) and *Openness in Cross-Cultural Engagement* (II-11). These results are consistent with the view that language education should avoid cultural essentialism and cultivate students' hybrid identity capabilities [28]. This equal emphasis addresses the limitations of earlier models that either overemphasized cultural loyalty or prioritized global perspectives at the expense of local roots [55]. Empirically, surveys revealed that students with imbalanced attitudes (e.g., excessive pride without openness) struggled to engage in meaningful cross-cultural dialogues [56], while those with balanced attitudes demonstrated better cultural empathy and communication adaptability [57]. This aligns with the sociopolitical goal of fostering an inclusive cultural identity—one that values local roots while remaining open to global perspectives—rather than promoting cultural superiority or dilution. Intercultural competence does not require replacing one's existing cultural identity. Rather, it involves expanding the range of identities one can engage with [5]. This perspective supports the equal emphasis placed on both cultural pride and openness in the current study.

In university EFL assessment, cultural pride can be evaluated through tasks such as the "Pride Portfolio" (II-10 in S5 Table). The task requires learners to select personally meaningful cultural artifacts and articulate their symbolic significance in English, fostering a personal connection to the symbols they will later mediate. This emphasis on the affective foundation for symbolic action represents an extension of Kramsch's model within an assessment framework. Cross-cultural openness can be measured through bidirectional comparison tasks (e.g., Dragon Boat Races vs. Christmas customs in bilingual archives), highlighting the importance of fostering critical thinking and cultural sensitivity in cross-cultural interactions [58].

This balanced assessment approach (cultural pride/openness both at 11.81%) embodies the CLIL integration paradigm, where language proficiency and critical cultural inquiry co-develop through scaffolded tasks. This approach not only enhances their language skills but also fosters a deeper engagement with cultural content and critical thinking about cultural comparisons, aligning with the principles of higher-order thinking (HOT) and interdisciplinary learning emphasized in contemporary EFL education [17,18].

## Cognitive dimension

The indicators under Local Cognition and Understanding provide the essential foundational knowledge of the symbolic systems learners will later mediate and manipulate in the Application dimension. However, a significant divergence in expert opinions was observed during the first Delphi round, particularly regarding abstract cultural indicators such as the *Local Political System* and *Local Economic System,* which yielded coefficients of variation (CV) of 0.45 and 0.36, respectively. These relatively high CV values suggest that experts questioned the relevance of such institutional topics in the context of EFL learning. This is consistent with the view that language learning should be combined with meaningful content and communicative practices, rather than just focusing on abstract cultural knowledge [59]. Consequently, by deleting or merging these abstract topics, the study ensures that the included cultural content is both comprehensible to university students and directly relevant to their language learning needs.

Within this dimension, three secondary indicators – *Local Cultural Practices, Art, and Literature* (II-2), *Daily Life Experiences* (II-4), and *National Virtues* (II-5) *and Qualities* – exhibited perfect evaluative consensus (CV = 0.00). This consensus confirms the critical role of culturally-embedded assessment tasks in measuring EFL learners' ability to mediate local knowledge through English, as exemplified by "a day in the life" documentation (II-4) and local figure narration (II-5) tasks in S5 Table. Aligned with the cognitive authenticity principle [2], these evaluation criteria demonstrate how contextually-grounded content enhances linguistic encoding and identity positioning in assessment outcomes. For instance, the cultural artifact transformation task (II-2 in S5 Table) evaluates symbolic mediation competence through a multidimensional framework assessing cultural authenticity, linguistic precision, and effective embodiment of cultural meaning. This multidimensional approach operationalizes Kramsch's symbolic competence framework, requiring learners to move beyond lexical equivalence and forge semiotic resonance between form and cultural substance.

The substantial weighting of *Local Development Achievements* underscores the evaluation framework's recognition of contemporary cultural production as equally significant as traditional manifestations. This indicatorputs symbolic competence into practice by requiring learners to interpret abstract cultural values through the concrete symbols of quantitative data and development narratives, mediating between different symbolic systems. In university assessment contexts, business students' cultural awareness is measured through tasks like creating "local development data visualization" reports in English (e.g., analyzing cultural industry GDP contributions). These tasks necessitate contextualizing technological/cultural narratives within specific development metrics [3], validating learners' ability to concretize socio-cultural abstractions through quantitative discourse.

Notably, the indicator *Local and Global Issues in Everyday Contexts* (M = 3.93, global weight = 0.0200), though scoring relatively low, was retained by experts for its role in cultivating critical global citizenship. In university EFL teaching, this is applied through graded tasks: advanced learners' critical analysis capabilities are measured through English debates on "local issues with global implications" (e.g., "ancient town tourism vs. cultural authenticity preservation"), while intermediate learners demonstrate their understanding through structured pro-con reports on topics like "dialect protection and tourism development." The tiered assessment design empirically substantiates the claim [26] that internationalized curricula preserve cultural heterogeneity through critical engagement. Simultaneously, the criteria implement the framework's conceptual synthesis of linguistic proficiency and intercultural consciousness—quantifying learners' competence in articulating local identities within global discursive paradigms.

## Integration with existing standard frameworks in university context

The practical application of this evaluation model can be aligned with widely recognized standard frameworks, such as the Common European Framework of Reference for Languages (CEFR) and the American Council on the Teaching of Foreign Languages (ACTFL) proficiency guidelines. Although this study highlights the inadequacies of current frameworks (such as CEFR and ACTFL) in assessing local cultural expression, the new model is not intended to replace these standardized systems but to achieve a synergistic enhancement through a complementary, hierarchical strategy.

The CEFR offers a comprehensive set of language proficiency descriptors spanning diverse proficiency levels [35]. We propose integrating the local cultural awareness assessment model with the CEFR starting from the intermediate level, employing a descriptor mapping approach. Specifically, this integration can be achieved by embedding local cultural indicators into the CEFR's mediational and plurilingual descriptors, as underscored in the document regarding "action-oriented scenarios" and "plurilingual repertoires" [35]. As outlined in S6 Table, this strategic embedding aligns secondary indicators with CEFR levels B1 to C2, illustrating how descriptors like "relaying specific information" (B2) and "facilitating pluricultural space" (C1) can be adapted to assess local cultural competence.

B1–B2 Intermediate Level Integration: At these levels, foundational local cultural indicators (e.g., II-1 Local History, II-4 Daily Life Experiences) can be integrated into CEFR's "Mediating Concepts" and "Reception of Information" descriptors. For instance, II-1's requirement for learners to "discuss local history events in English" aligns with CEFR B2's descriptor for "relaying specific information" [35], where learners are expected to "summarize historical contexts with cultural significance". Similarly, II-6 (Local Ethical and Legal Systems) can be grafted onto CEFR's B2 "Explaining Data" scale, as the document highlights the importance of mediating social norms through language.

C1–C2 Advanced Level Integration: At higher proficiency tiers, sophisticated cultural indicators (e.g., II-15 Local Image Promotion, II-18 Cultural Comparison) map to CEFR's "Plurilingual Mediation" and "Facilitating Pluricultural Space" domains. For example, II-18's requirement to "compare local and target cultures" directly aligns with CEFR C1's descriptor for "mediating cross-cultural dialogue", where learners are expected to "negotiate cultural differences in complex scenarios". Additionally, II-15's focus on "promoting local identity in English" resonates with the emphasis of CEFR on "activating plurilingual repertoires for social agency", which can be embedded in CEFR C2's "Strategic Mediation" scale.

Cross-Level Task Design for Cultural Embedding: The document's advocacy for "scenario-based assessment" provides a framework for designing tasks that bridge proficiency levels [35]. For instance, in B1–B2 tasks, learners could complete a "local food culture interview" (II-4) using CEFR B2's "Oral Interaction" descriptors, which emphasize "describing daily practices with cultural relevance". For C1–C2 tasks, a "local heritage preservation campaign" (II-15) could be evaluated against CEFR C1's "Mediation of Public Discourse" criteria, requiring learners to "craft persuasive messages that integrate local values".

Similarly, ACTFL emphasizes "intercultural communicative competence," focusing on the appropriate use of language in real social and cultural contexts [36]. We propose using the evaluation indicators established in this study to form a dual-track assessment mechanism (Fig 5). For example, a cultural awareness dimension can be added to the ACTFL Oral Proficiency Interview (OPI), allowing independent scoring of language fluency and cultural mediation while sharing weights (suggested ratio: 70% language + 30% culture). This integrated solution, which balances the emphasis on language and cultural competence, retains the reliability and validity of standardized tests and addresses the missing identity expression through the cultural competence matrix (S7 Table). This approach responds to the call for a "mixed assessment method" (quantitative + qualitative) to measure cultural depth [38]. Meanwhile, the dual-track model proposed in this study requires minimal modification to the existing ACTFL protocols but significantly enhances ecological validity in multilingual contexts.

## Limitations and future directions

This study is a preliminary exploration to evaluate local cultural awareness among EFL learners using Delphi and AHP methodologies. Although the panel of 15 Chinese experts provides valuable insights, its homogeneous composition is a significant limitation. Their shared background may constrain the cross-cultural applicability of the index system. Furthermore, the lack of input from key stakeholders—such as EFL learners, employers, and educators from diverse educational systems—restricts our understanding of the framework's practical implementation across user groups.

Building on this foundation, subsequent studies should pursue two complementary paths. First, cross-cultural validation in non-Chinese settings would involve not only assessing contextual adjustments but also conducting empirical pilot tests (e.g., pre/post-intervention surveys measuring learners' intercultural competence) to quantify the framework's effectiveness across diverse EFL contexts. This line of research would thereby enrich the framework by incorporating global perspectives on how to balance local and global cultural elements in curricula. Second, longitudinal investigations are needed to examine the long-term impact of local cultural awareness on students' language development and intercultural competence. Such research must employ mixed-methods designs, triangulating quantitative data (e.g., from standardized tests) with qualitative evidence (e.g., from learner diaries or case studies) to capture nuanced insights into how learners navigate cultural hybridity across different linguistic environments.

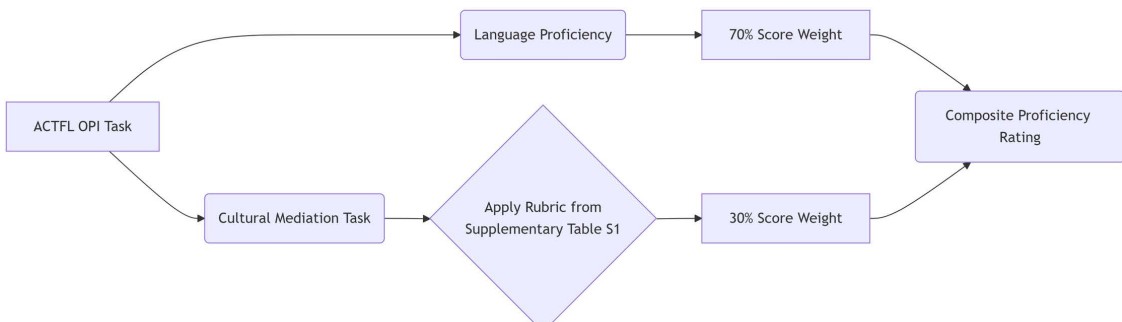

**Fig 5. Schematic of the dual – track assessment process for the ACTFL OPI task.**

Through acknowledging the potential for broader application and inviting interdisciplinary dialogue, future studies can refine the framework to accommodate the diverse needs of EFL learners worldwide without sacrificing sensitivity to regional cultural identities. This approach ultimately ensures that educational interventions can both honor local voices and foster global communicative agility.

## Conclusion

Translating Kramsch's symbolic competence theory into quantifiable assessment metrics, this study constructs a comprehensive evaluation framework for local cultural awareness in EFL education through the Delphi-AHP methodology. The empirically validated framework comprises three core dimensions—Local Expression and Application (45.27%), Local Affective Attitudes (33.01%), and Local Cognition and Understanding (21.72%)— alongside 18 secondary indicators, such as "Cultural Comparison" and "Local Image Promotion". Its primary innovation lies in bridging sociocultural theories with classroom practice by operationalizing abstract constructs into AHP-weighted evaluation criteria, thereby addressing the overemphasis on Anglophone content in standardized curricula.

The framework offers versatile applications: it guides curriculum developers in benchmarking cultural balance, enables AI-driven adaptive learning through metrics like "Cultural Pride", and provides policymakers with tools to assess glocal citizenship cultivation. By emphasizing local storytelling and cross-cultural comparison, the framework empowers learners as cultural agents while fostering objective cultural understanding. This research establishes a foundational model for integrating local cultural content in EFL education and paves the way for future cross-cultural validation and longitudinal impact studies.

## Supporting information

**S1 Fig. Hierarchical Model of EFL Learners' Local Cultural Awareness Evaluation System.**
(TIFF)

**S1 Table. First-round Delphi consultation on indicators of the local cultural awareness evaluation system for EFL learners.**
(DOCX)

**S2 Table. Second-round Delphi consultation on secondary indicators of the local cultural awareness evaluation system for EFL learners.**
(DOCX)

**S3 Table. Pairwise comparison matrix for primary indicators.**
(XLSX)

**S4 Table. Pairwise comparison matrix for secondary indicators.**
(XLSX)

**S5 Table. Operationalization framework for local cultural awareness assessment.**
(DOCX)

**S6 Table. Index system with CEFR alignment.**
(DOCX)

**S7 Table. ACTFL-OPI Integration.**
(DOCX)

## Author contributions

**Conceptualization:** Lunan Fan.

**Data curation:** Lunan Fan.

**Formal analysis:** Lunan Fan.

**Funding acquisition:** Lunan Fan.

**Investigation:** Lunan Fan, Guiping Cheng, Na Yan.

**Methodology:** Lunan Fan, Guiping Cheng.

**Project administration:** Lunan Fan.

**Resources:** Lunan Fan.

**Software:** Lunan Fan.

**Supervision:** Lunan Fan.

**Validation:** Lunan Fan.

**Visualization:** Lunan Fan, Na Yan.

**Writing – original draft:** Lunan Fan.

**Writing – review & editing:** Lunan Fan, Guiping Cheng.

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
