## [Decision Letter · Decision Letter 0]

13 May 2025

Dear Dr. Fan,

Thank you for submitting your manuscript to PLOS ONE. After careful consideration, we feel that it has merit but does not fully meet PLOS ONE’s publication criteria as it currently stands. Therefore, we invite you to submit a revised version of the manuscript that addresses the points raised during the review process.

We look forward to receiving your revised manuscript.

Kind regards,

Muhammad Zammad Aslam

Academic Editor

PLOS ONE

Journal Requirements:

The initials of the authors who received each award: F.L.

This study was supported by the "2023 Guizhou Provincial Education Science Planning Project and Yue-Qian Special Project" from the Guizhou Provincial Department of Education. The project is titled "Constructing National Identity in College English through the Integration of Guizhou's Local Cultural Resources" and has been awarded the project number: 2023B035.

For further information regarding the funding support, please refer to the following links:

Guizhou Provincial Department of Education website: https://jyt.guizhou.gov.cn/index.html

Announcement of the project establishment: https://jyt.guizhou.gov.cn/xwzx/tzgg/202311/t20231113_83074270.html

The funding body was primarily involved in the release of the project guidelines and the selection of high-quality projects during the establishment process.

4. We notice that your supplementary [figures] are included in the manuscript file. Please remove them and upload them with the file type 'Supporting Information'. Please ensure that each Supporting Information file has a legend listed in the manuscript after the references list.

5. Please remove all personal information, ensure that the data shared are in accordance with participant consent, and re-upload a fully anonymized data set.

Additional Editor Comments :

Recommendations for Major Revisions

1. Generalizability of sample

- Please provide justifications for your sample limitations, explicitly acknowledging the validations discussed in your discussion and explaining future testing to make your framework universal across diverse EFL contexts (e.g., Latin America, Europe, US, etc.).

2. Indicator Operationalization

- Please provide concrete examples for secondary indicators, such as Local storytelling and Cultural Pride, to illustrate practical implications. This can be achieved by detailing scenarios for hypothetical assessment in a supplementary table that explains future sample assessment tasks or rubrics.

3. Theoretical Issues

- Please explain your models more within your local context, such as ACTFL and CEFR, by creating an integrated model of local culture descriptors. This would also aid in proposing a model for practitioners based on the implications of your study.

4. Data Transparency Explained

- Your data is not fully present in the manuscript. Please ensure that you submit your data to a repository in accordance with the nature of your study.

5. Ethical Considerations

- Are others’ identities kept secured? Please explain it in the EC section

6. Theoretical Integration

- Add recent citations for clear integration, as suggested by the Reviewer, for bridging theory and practice—specifically in the discussion section under the heading “cognitive processes and feedback preferences."

7. Figures

- Please use good software to improve your figures

8. Discussion/Conclusion

- Please write your specific implications

- Learners’ narratives are not discussed properly in comparison with existing studies. Compare local culture to international cultures

- Include DOIs for all references.

- Professional proofreading is needed

Reviewers' comments:

Reviewer's Responses to Questions

**Comments to the Author**

1. Is the manuscript technically sound, and do the data support the conclusions?

Reviewer #1: Yes

Reviewer #2: Partly

2. Has the statistical analysis been performed appropriately and rigorously?

Reviewer #1: Yes

Reviewer #2: Yes

3. Have the authors made all data underlying the findings in their manuscript fully available?

Reviewer #1: Yes

Reviewer #2: No

4. Is the manuscript presented in an intelligible fashion and written in standard English?

Reviewer #1: Yes

Reviewer #2: Yes

Reviewer #1: 1. The paper is interesting as it offers a framework that can be expanded and applied in other contexts too.

2. Improve the aim of the study. Is it to use Delphi, or to examine matters at hand through Delphi?

3. Revise the format of the article. The colour scheme for the figures may not be the most effective.

Reviewer #2: The authors have done great job to compose a well comprehend article, however some major changes would help to improve the integrity and academic value of the study. I would suggest authors to consider recommended comments provided by attached file.

**Do you want your identity to be public for this peer review?** For information about this choice, including consent withdrawal, please see our Privacy Policy

Reviewer #1: **Yes: ** Shaidatul Akma Adi Kasuma

Reviewer #2: No

---

## [Author Response · Author response to Decision Letter 1]

27 Jun 2025

Dear Reviewer,

I hope this message finds you well. I am writing to express my sincere gratitude for the thoughtful comments and constructive feedback you provided on my manuscript. Your insights have been instrumental in helping me refine the work, and I believe they will significantly enhance the quality of the submission.

All my responses to your specific comments are detailed in the attached document. I’ve strived to address each point thoroughly, and I hope it demonstrates my commitment to improving the manuscript based on your valuable input.

Thank you again for your time, expertise, and dedication to this review process. It is because of reviewers like you that scholarly work can progress and mature.

Best regards,

Fan Lunan

---

## [Decision Letter · Decision Letter 1]

28 Jul 2025

Dear Dr. Fan,

Thank you for submitting your manuscript to PLOS ONE. After careful consideration, we feel that it has merit but does not fully meet PLOS ONE’s publication criteria as it currently stands. Therefore, we invite you to submit a revised version of the manuscript that addresses the points raised during the review process.

We look forward to receiving your revised manuscript.

Kind regards,

Muhammad Zammad Aslam, Ph.D.

Academic Editor

PLOS ONE

Journal Requirements:

Reviewers' comments:

Reviewer's Responses to Questions

**Comments to the Author**

Reviewer #1: All comments have been addressed

Reviewer #2: (No Response)

2. Is the manuscript technically sound, and do the data support the conclusions?

Reviewer #1: Yes

Reviewer #2: Partly

3. Has the statistical analysis been performed appropriately and rigorously?

Reviewer #1: Yes

Reviewer #2: Yes

4. Have the authors made all data underlying the findings in their manuscript fully available?

Reviewer #1: Yes

Reviewer #2: Yes

5. Is the manuscript presented in an intelligible fashion and written in standard English?

Reviewer #1: Yes

Reviewer #2: Yes

Reviewer #1: The article is well-written and presented. The data is rigourous. There are several parts that may require further clarity as highlighted in the manuscript.

Reviewer #2: 1,The manuscript is excessively long, with numerous repetitive examples, citations, and table explanations (e.g., extensive elaboration on indicators that repeat the same construct wording). The Results section, while data-rich, suffers from redundancy between narrative and table content. Condense repeated discussions on Delphi/AHP procedures. Consider moving detailed indicator breakdowns to appendices or supplementary files.

2,Kramsch and Byram are central, but no conceptual model or diagram links the three dimensions (cognition, affect, behavior) to symbolic competence. While symbolic competence is mentioned, symbolism as a construct is underdeveloped. Add a conceptual model figure showing how symbolic competence maps onto your three dimensions. Also, better define how your index extends vs. operationalizes Kramsch’s theory.

3,All 15 experts are from mainland China, and most have similar academic backgrounds. This weakens the cross-cultural applicability of the index. No diversity in learner or stakeholder perspectives (e.g., learners, employers, teachers from different systems). Acknowledge this limitation explicitly in the Discussion. Consider a future validation phase in diverse contexts (e.g., South Asia, ASEAN) as a next step.

4,While AHP calculations are provided, rationale for certain weightings lacks theoretical grounding. For instance, why is storytelling (13.67%) weighted more heavily than historical understanding (3.35%)? The decision to equal-weight affective attitudes like “pride” and “critical openness” is sociopolitically sensitive—more explanation is needed.

Justify key weight assignments with either empirical pilot results or pedagogical priorities tied to established models (e.g., Bloom’s, Vygotsky).

5,Numerous grammatical issues, unclear sentence structures, and mixed citation styles (APA, numeric). Some sections (e.g., “Second Delphi survey results”) read more like reports than academic narrative. Strong language editing is necessary. Use consistent academic English, e.g., replace "improved indicators" with "refined indicators."

**Do you want your identity to be public for this peer review?** For information about this choice, including consent withdrawal, please see our Privacy Policy

Reviewer #1: **Yes: ** Shaidatul Akma Adi Kasuma

Reviewer #2: **Yes: ** Ushba Rasool

---

## [Author Response · Author response to Decision Letter 2]

24 Aug 2025

Dear Reviewers,

Thank you very much for your time and constructive comments. We have carefully addressed all of your points in the response letter and revised the manuscript accordingly. We believe the manuscript has been significantly improved as a result.

Sincerely,

The Authors

---

## [Decision Letter · Decision Letter 2]

28 Aug 2025

Assessing local cultural awareness in university EFL learners: A Delphi and AHP-based index framework

PONE-D-25-18530R2

Dear Dr. Fan,

We’re pleased to inform you that your manuscript has been judged scientifically suitable for publication and will be formally accepted for publication once it meets all outstanding technical requirements.

Kind regards,

Muhammad Zammad Aslam, Ph.D.

Academic Editor

PLOS ONE

Additional Editor Comments (optional):

Reviewers' comments:

Reviewer's Responses to Questions

**Comments to the Author**

Reviewer #2: All comments have been addressed

2. Is the manuscript technically sound, and do the data support the conclusions?

Reviewer #2: Yes

3. Has the statistical analysis been performed appropriately and rigorously?

Reviewer #2: Yes

4. Have the authors made all data underlying the findings in their manuscript fully available?

Reviewer #2: Yes

5. Is the manuscript presented in an intelligible fashion and written in standard English?

Reviewer #2: (No Response)

Reviewer #2: The authors have thoroughly revised the manuscript as suggested and i appreciate the effort made by author's in this process.

**Do you want your identity to be public for this peer review?** For information about this choice, including consent withdrawal, please see our Privacy Policy

Reviewer #2: **Yes: ** Ushba Rasool

---

## [Editor Report · Acceptance letter]

PONE-D-25-18530R2

PLOS ONE

Dear Dr. Fan,

I'm pleased to inform you that your manuscript has been deemed suitable for publication in PLOS ONE. Congratulations! Your manuscript is now being handed over to our production team.

Kind regards,

on behalf of

Dr. Muhammad Zammad Aslam

Academic Editor

PLOS ONE